# Heteromerization of Endogenous Mu and Delta Opioid Receptors Induces Ligand-Selective Co-Targeting to Lysosomes

**DOI:** 10.3390/molecules25194493

**Published:** 2020-09-30

**Authors:** Lyes Derouiche, Florian Pierre, Stéphane Doridot, Stéphane Ory, Dominique Massotte

**Affiliations:** 1French National Centre for Scientific Research, Institut des Neurosciences Cellulaires et Intégratives, University of Strasbourg, 67000 Strasbourg, France; lyes.derouiche@gmail.com (L.D.); florianb.pierre@gmail.com (F.P.); ory@inci-cnrs.unistra.fr (S.O.); 2French National Centre for Scientific Research, Chronobiotron, 67200 Strasbourg, France; doridot@inci-cnrs.unistra.fr

**Keywords:** mu opioid receptor, delta opioid receptor, heteromer, internalization, primary hippocampal culture, lysosomes

## Abstract

Increasing evidence indicates that native mu and delta opioid receptors can associate to form heteromers in discrete brain neuronal circuits. However, little is known about their signaling and trafficking. Using double-fluorescent knock-in mice, we investigated the impact of neuronal co-expression on the internalization profile of mu and delta opioid receptors in primary hippocampal cultures. We established ligand selective mu–delta co-internalization upon activation by 1-[[4-(acetylamino)phenyl]methyl]-4-(2-phenylethyl)-4-piperidinecarboxylic acid, ethyl ester (CYM51010), [d-Ala2, NMe-Phe4, Gly-ol5]enkephalin (DAMGO), and deltorphin II, but not (+)-4-[(α*R*)-α-((2*S*,5*R*)-4-Allyl-2,5-dimethyl-1-piperazinyl)-3-methoxybenzyl]-*N*,*N*-diethylbenzamide (SNC80), morphine, or methadone. Co-internalization was driven by the delta opioid receptor, required an active conformation of both receptors, and led to sorting to the lysosomal compartment. Altogether, our data indicate that mu–delta co-expression, likely through heteromerization, alters the intracellular fate of the mu opioid receptor, which provides a way to fine-tune mu opioid receptor signaling. It also represents an interesting emerging concept for the development of novel therapeutic drugs and strategies.

## 1. Introduction

The opioid system modulates a large number of functions including nociception, emotional responses, reward and motivation, and cognition, as well as neuroendocrine physiology and autonomic functions [1,2]. It is composed of three G-protein-coupled receptors, mu, delta, and kappa, and three families of opioid peptides, the enkephalins, dynorphins, and endorphins [3]. Several decades of pharmacology have uncovered the complexity of the opioid pharmacology and evidenced functional interactions between receptors that can take place at different levels, including within the cell [4,5]. This led to postulate the formation of functional association between different opioid receptor types to generate a novel entity with specific pharmacological, signaling, and trafficking properties called heteromers [6]. Heteromers within the opioid family were postulated for the first time about 20 years ago involving the delta and the kappa opioid receptors [7]. Heteromerization of mu and delta opioid receptors was then proposed shortly after [8] and extensively studied in co-transfected cells [9]. Mu and delta opioid receptors have different intracellular fate when internalized, with mu opioid receptors being recycled quickly to the plasma membrane [10,11] and delta opioid receptors being degraded in the lysosomal compartment [12,13,14]. In co-transfected HEK293 cells, co-internalization of mu and delta opioid receptors was reported following activation by the mu agonists [d-Ala^2^, NMe-Phe^4^, Gly-ol^5^]enkephalin (DAMGO) [15,16,17,18] or methadone [19] or following activation by the delta agonists SNC80 [17,18], [*H*-Dmt-Tic-NH-CH(CH2-COOH)-Bid] (UFP512) [17], deltorphin I [18], deltorphin II [16,17,18], or d-Pen2, d-Pen5 -enkephalin (DPDPE) [17]. Co-targeting to the lysosomal compartment was observed following activation by deltorphin I [18] or methadone [19]. However, differences in the cellular content are known to exist between cell types that may impact receptor functioning [20,21] and underline the need for studies on endogenous receptors. Although a previous report indicated that the mu agonist DAMGO induced co-internalization and co-recycling of mu and delta opioid receptors in Dorsal root ganglia (DRG) cultures pretreated with morphine, suggesting that mu–delta heteromerization may affect the trafficking of the delta opioid receptor in these conditions [22], little is known so far regarding the consequences the trafficking of mu–delta heteromers in neurons.

Using double-fluorescent knock-in mice co-expressing functional mu and delta opioid receptors respectively fused to the red fluorescent protein mCherry or the green fluorescent protein eGFP, we previously mapped neurons co-expressing mu and delta opioid receptors [11]. In the hippocampus, they corresponded to γ-aminobutyric acid (GABA) interneurons with 70% being parvalbumin-positive [23]. We also established close physical proximity of the two receptors in the hippocampus, a prerequisite to mu–delta heteromerization [11]. Here, we took advantage of the double-fluorescent knock-in mice to examine whether mu–delta physical proximity was also associated with functional changes by monitoring mu and delta receptor internalization in primary hippocampal cultures. We showed ligand-specific mu–delta receptor co-internalization induced by the mu–delta-biased agonist CYM51010 [24,25], the mu agonist DAMGO, and the delta agonist deltorphin II, but not the mu agonists morphine and methadone or the delta agonist SNC80. We also established the sorting of mu–delta heteromers to the lysosomal compartment indicating that mu–delta heteromerization affects the intracellular fate of the mu opioid receptor in its native environment. These data point to mu–delta heteromerization as a means to fine-tune mu opioid receptor signaling and neuronal activity.

## 2. Results

### 2.1. Endogenous Mu–Delta Heteromers Are Present at the Neuronal Surface under Basal Conditions

In agreement with our previous reports using the fluorescent knock-in mice expressing delta-eGFP and/or mu-mCherry [11,13,26,27], both mu and delta opioid receptors were detected at the plasma membrane in primary hippocampal neurons under basal conditions (Figure 1A). Quantification of the receptor density using the ICY bioimaging software [28] indicated that the fluorescence density at the cell surface was 2.5-fold higher compared to the cytoplasm for either receptor (Figure 1B). Merged images highlighted an overlay of the green and red fluorescence at the surface of the neuron, and quantification of the density of receptor co-localization indicated higher co-localization at the plasma membrane compared to the cytoplasm (Figure 1C), with only 10% of the receptors co-localized in the cytoplasm (Figure 1D).

Our data, thus, indicate close physical proximity of endogenous mu and delta opioid receptors at the plasma membrane of hippocampal neurons and suggest constitutive mu–delta heteromerization at the surface of neurons.

### 2.2. CYM51010 Induces Mu–Delta Receptor Co-Internalization and Co-Localization in the Late Endosomal Compartment in Primary Hippocampal Cultures

CYM51010 was reported as a mu–delta-biased agonist because its antinociceptive effect was blocked by an antibody selective for mu–delta heteromers and its activity was reduced in mice deficient for the mu or delta opioid receptor [24,25]. We, therefore, tested whether activation by this ligand (concentration range 10 nM to 10 μM) triggered mu and delta receptor internalization in primary hippocampal cultures from double-fluorescent knock-in mice. CYM51010 concentrations equivalent to, or higher than, 400 nM induced mu-mCherry and delta-eGFP internalization as seen from the decrease in fluorescence density associated with the plasma membrane and the appearance of fluorescent intracellular vesicles (Figure 1A,B). Quantification of the extent of co-localization 15, 30, and 60 min after agonist administration showed that the fraction of mu and delta opioid receptors that co-localized at the plasma membrane significantly decreased (Figure 1C), whereas mu–delta receptor co-localization increased in the cytoplasm at the three time points (Figure 1D), establishing co-internalization of the receptors. Triple immunofluorescence labeling with Lysosomal-associated membrane protein 1 (LAMP1) as a marker of the late endosomal–lysosomal compartment showed increased co-localization with mu-mCherry and delta-eGFP 60 min after activation by CYM51010 (Figure 2), suggesting that mu and delta opioid receptors are targeted together to the degradation pathway.

We then sought to investigate whether internalization of the mu opioid receptor by CYM51010 was promoted by its association with the delta opioid receptor. In primary hippocampal cultures from single fluorescent knock-in animals expressing mu-mCherry and deficient for the delta opioid receptor, CYM51010 concentrations up to 1 μM failed to induce mu-mCherry internalization (Figure 3A,B) with only limited mu opioid receptor clustering and subcellular redistribution at 10 μM (Figure 3B).

We also examined whether internalization of the delta opioid receptor upon activation by CYM51010 required mu opioid receptor co-expression. In primary hippocampal cultures from single fluorescent knock-in animals expressing delta-eGFP and deficient for the mu opioid receptor, CYM51010 (400 nM) induced internalization of the delta opioid receptor (Figure 3C). In addition, predominant intracellular localization was observed 30 and 60 min after agonist application (Figure 3D) in agreement with kinetics described for the delta selective agonist SNC80 [12,13,26], indicating that CYM51010 was able to promote delta opioid receptor internalization despite the lack of mu opioid receptor expression.

Together, these data establish that mu opioid receptor internalization by CYM51010 is dependent on mu–delta receptor co-expression and directs the mu opioid receptor to the late endocytic compartment.

### 2.3. CYM51010-Induced Mu–Delta Receptor Co-Internalization Is Blocked by Pretreatment with Mu- or Delta-Selective Antagonists

In neurons expressing one receptor only, CYM51010 activation led to the internalization of delta but not mu opioid receptors (Figure 3). We, therefore, sought to determine whether co-internalization by CYM51010 required the two receptors to be in an active conformation. To this aim, we examined the impact of pretreatment for 15 min with the mu-selective antagonists beta-funaltrexamine (β-FNA) (20 nM) orCTAP (200 nM). Both antagonists prevented mu opioid receptor cellular redistribution but did not block delta opioid receptor internalization (Figure 4A,C,D). These results suggest that an active conformation of the mu opioid receptor is required for mu–delta co-internalization.

We evaluated the need for delta opioid receptor activation in the co-internalization process. Whereas pretreatment with mu antagonists blocked mu but not delta opioid receptor internalization, pretreatment with the selective delta antagonists naltrindole or tic-deltorphin 200 nM blocked the internalization of both delta and mu opioid receptors (Figure 4B–D). This indicates that mu–delta cellular redistribution is driven by delta opioid receptor expression and activation.

Together, this result indicates that mu–delta receptor co-internalization upon CYM51010 activation is driven by delta opioid receptors and requires both mu and delta opioid receptors to be in an active conformation.

### 2.4. Mu–Delta Receptor Co-Internalization Is Ligand-Specific

We examined whether other synthetic opioid agonists were able to promote mu–delta receptor co-internalization in primary hippocampal cultures. Mu–delta receptor co-localization in the cytoplasm was increased 30 min after stimulation with the mu agonist DAMGO (1 μM) or the delta agonist deltorphin II (100 nM), but not upon stimulation with the delta agonist SNC 80 (100 nM) or the mu agonists morphine (10 μM) or methadone (1 μM) (Figure 5). These data establish ligand-specific internalization of endogenous mu–delta heteromers by exogenous opioids.

## 3. Discussion

In this study, we used primary hippocampal cultures of double-fluorescent knock-in mice to investigate the impact of neuronal co-expression on the internalization of native mu and delta opioid receptors.

### 3.1. Mu–Delta Co-Internalization Is Induced by Different Ligands in Native or Co-Transfected Cells

We showed here that co-internalization of endogenous mu and delta opioid receptors was ligand-dependent and took place following activation by the mu–delta-biased agonist CYM51010, the mu agonist DAMGO, or the delta agonist deltorphin II, but not following activation by the mu agonist morphine. This is consistent with previous reports in co-transfected HEK293 cells in which co-internalization of the receptors was promoted by DAMGO [15,16,17,18] or deltorphin II [16,17,18], but not morphine [19]. No co-internalization of mu and delta opioid receptors was observed following activation by the delta agonist SNC80, in agreement with the absence of receptor co-internalization in the spinal cord of delta-eGFP knock-in mice following SNC80 (10 mg/kg intraperitoneal (i.p.)) administration [29]. These results obtained in native environment are, however, in marked contrast to the reported mu–delta co-internalization in co-transfected cells [17,18]. Furthermore, we did not evidence mu–delta co-internalization by the mu agonist methadone although this ligand induced mu–delta co-trafficking in in co-transfected cells [19]. Collectively, these observations highlight the difficulty to draw definite conclusions from data collected in heterologous systems. Differences between native and heterologous environments may reflect distinct cellular contents [20]. Internalization of the delta opioid receptor was dependent on G protein coupled receptor kinase 2 (GRK2) in cortical neurons but not in transfected HEK293 cells, although the latter expressed GRK2 and supported GRK2-mediated internalization of other GPCRs [30]. Similarly, the ability of ligands to differentially activate signaling pathways in AtT20 neuroblastoma and CHO cell lines uncovered clear influence of the cellular background on mu opioid receptor signaling [31]. Expression of high levels of receptors in a non-native environment can also artificially elicit interactions that would not occur in vivo and could subsequently affect functional responses. Accordingly, low levels of mu (10–15 fmol/mg protein) [32,33] and delta opioid receptors (30–50 fmol/mg protein) [32,33,34,35] are present in the mouse hippocampus, whereas heterologous receptors are most often expressed in the picomolar range.

A potential influence resulting from the C-terminal fusion to a fluorescent protein is also to be considered. Addition of the fluorescent tag did not modify the expression level of the mu opioid receptor [11] but induced a twofold increase in delta opioid receptor expression [12]. In particular, strong surface expression of the delta-eGFP construct in the hippocampus could alter receptor trafficking and signaling. However, no overt change in the neuroanatomical distribution, pharmacological, and signaling properties or behavioral response has been evidenced so far in the knock-in mice expressing the delta-eGFP and/or mu-mcherry fluorescent fusions (reviewed in [36]). Importantly, delta-eGFP surface expression varies across the nervous system and is increased upon chronic morphine administration [27] or in neuropathic pain conditions [37], as previously reported for wild-type receptors (reviewed in [5,36]). Moreover, the use of the delta-eGFP fusion enabled detecting in vivo partial receptor internalization in response to a physiological stimulation [26] or upregulation following Pavlovian training [38]. The fluorescent knock-in mice, therefore, appear to be well-suited reporters for native opioid receptor studies.

### 3.2. CYM51010 Activation Induces Co-Targeting of Mu and Delta Receptors to the Lysosomal Compartment

Native mu opioid receptors rapidly recycle back to the plasma membrane [10,11], whereas native delta opioid receptors are slow-recycling receptors that are degraded in the lysosomal compartments [13]. Here, we observed that native mu–delta heteromers were targeted to the lysosomal compartment in primary neurons triggering a change in the mu opioid receptor intracellular fate following activation by CYM51010. Although CYM51010 was reported as a mu–delta heteromer biased agonist [24], it also binds to mu or delta opioid receptors expressed alone. Previous reports suggested that CYM51010 would not activate delta opioid receptors because [^35^S]GTPγS activation by CYM51010 was not prevented by antibodies specific for the delta opioid receptor [24], and CYM51010 administration did not modify mechanical or thermal allodynia in neuropathic mu knockout animals [25]. However, our data revealed that CYM51010 promoted delta opioid receptor internalization in mu knockout mice indicating that it could activate this receptor in the absence of the mu opioid receptor. CYM51010 could also activate mu opioid receptors because antibodies specific for the mu opioid receptor reduced [^35^S]GTPγS activation by CYM51010, although to a lesser extent than mu–delta-specific antibodies [24]. Moreover, CYM51010 had analgesic properties in delta knockout mice [25]. As shown here, activation of the mu opioid receptor by CYM51010 was not associated with internalization when the receptor was expressed alone. On the other hand, CYM51010 induced internalization of the mu opioid receptor when associated with the delta opioid receptor, suggesting that its binding triggered a different conformation of the mu opioid receptor that allowed beta-arrestin recruitment. Mu-selective (CTAP, cyprodime, β-FNA) or delta-selective (naltrindole, tic-deltorphin) antagonists prevented endogenous mu–delta co-internalization in agreement with previously reported inhibition of the mu or delta receptor, respectively, by a delta or a mu selective antagonist in co-transfected HEK293 cells [17]. These observations strongly suggest that co-trafficking requires both receptors in an active conformation and that mu–delta co-internalization did not result from random nonfunctional contacts elicited by receptor close proximity in the membrane.

Interestingly, delta antagonists blocked mu–delta receptor co-internalization, whereas mu antagonists only blocked mu opioid receptor internalization without affecting delta opioid receptor internalization and degradation. These data indicate that co-internalization was driven by the delta opioid receptor, possibly through constitutive β-arrestin recruitment [39,40]. Delta antagonists could, therefore, inhibit co-sequestration of the receptors by disrupting contacts between delta opioid receptors and β-arrestins, which would in turn destabilize the interface between the mu and delta opioid receptors.

### 3.3. Mu and Delta Opioid Receptors Form Functional Heteromers in the Hippocampus

Physical proximity of the receptors was established by co-immunoprecipitation in the hippocampus, where neuronal co-expression of mu and delta opioid receptors was mostly detected in parvalbumin-positive neurons [11]. Here, we confirmed co-expression of the two receptors at the plasma membrane in basal conditions. We also established that CYM51010, an agonist that preferentially binds mu–delta heteromers [24], induced co-internalization of the two receptors. In addition, co-internalization changed the intracellular fate of the mu opioid receptor compared to neurons where the receptor was expressed alone. Indeed, the mu opioid receptor was targeted to the lysosomal compartment instead of being recycled to the plasma membrane. Altogether, these observations satisfy the criteria for receptor heteromerization as defined by the International Union of Basic and Clinical Pharmacology (IUPHAR) [41] and establish the presence of functional mu–delta heteromers in hippocampal neurons.

## 4. Materials and Methods

### 4.1. Animals

Double knock-in mice co-expressing fluorescent mu and delta opioid receptors (mu-mCherry/delta-eGFP) were obtained by crossing previously generated single fluorescent knock-in mice expressing delta-eGFP or mu-mCherry, as described previously [11]. Single-fluorescent knock-in mice deficient for the other receptor were generated by crossing delta-eGFP with mu-knockout mice or mu-mCherry with delta knockout mice. The genetic background of all animals was 50:50 C57BL6/J:129svPas. Male and female adult mice (8–12 weeks old) were used for in vivo experiments.

Mice were housed in an animal facility under controlled temperature (21 ± 2 °C) and humidity (45% ± 5%) under a 12 h/12 h dark–light cycle with food and water ad libitum. All experiments were performed in agreement with the European legislation (directive 2010/63/EU acting on protection of laboratory animals) and received agreement from the French ministry (APAFIS 20 1503041113547 *(APAFIS#300).02)*.

### 4.2. Drugs


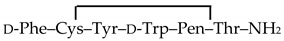
 (CTAP) (C-6352), beta-funaltrexamine (β-FNA) (O-003), fentanyl citrate (F3886), naltrindole (N-2893), [d-Ala^2^, NMe-Phe^4^, Gly-ol^5^]enkephalin (DAMGO) (E-7384), and deltorphin II (T-0658) were purchased from Sigma. (+)-4-[(α*R*)-α-((2*S*,5*R*)-4-Allyl-2,5-dimethyl-1-piperazinyl)-3-methoxybenzyl]-*N*,*N*-diethylbenzamide (SNC80) (cat n° 0764) was obtained from Tocris bioscience, 1-[[4-(acetylamino)phenyl]methyl]-4-(2-phenylethyl)-4-piperidinecarboxylic acid, ethyl ester (CYM51010) (ML-335) was obtained from Cayman chemical, and tic-deltorphin was synthesized as reported in [42]. Morphine hydrochloride was from Francopia, and methadone (M-0267) was from Sigma.

### 4.3. Primary Neuronal Culture

Primary neuronal cultures were performed as previously described [28]. Briefly, P0–P3 mice pups were decapitated, and their hippocampi were dissected and digested with papain (20 U/mL, Worthington cat. no. LS003126). Cells were plated (8–10 × 10^4^ cells/well) on polylysine (PLL, Sigma)-coated coverslips in 24-well plates. Cultures were maintained for 15 days in vitro (DIV) with half of the medium (Neurobasal A medium supplemented with 2% B27 (GIBCO, cat. no. 17504044), 2 mM glutamax (GIBCO, cat. no. 35050061), 0.5 mM glutamine and penicillin/streptomycin) changed every 5–7 days. Fully matured primary neurons (DIV 10 to 14) were used for all studies.

### 4.4. Drug Administration and Sample Preparation

DAMGO, naltrindole, CTAP, deltorphin II, morphine, methadone, and tic-deltorphin were dissolved in sterile milliQ water, CYM51010 was dissolved in saline solution with Dimethyl sulfoxide (DMSO) (0.2% final volume) and Tween-80 (1% final volume), and SNC 80 was dissolved in DMSO at 10 mg/mL. Drugs were added to the culture medium of mature neurons (as 1% of the total culture volume) (12–15 days in vitro) and incubated at 37 °C as indicated. Antagonists were added to the culture medium 15 min before agonist treatment.

For immunofluorescence studies, cultures were washed in cold 0.1 M phosphate-buffered saline pH 7.4 (PBS) and fixed with 4% paraformaldehyde in PBS. Cells were washed three times with cold PBS and kept at 4 °C until processing.

### 4.5. Fluorescent Detection with Antibodies

Primary neuronal cultures or brain sections were incubated in the blocking solution PBST (PBS with 0.2% Tween-20 (Sigma)) and 5% normal goat serum (Sigma)) for 1 h at room temperature (20–22 °C) and then overnight at 4 °C in the blocking solution with chicken anti-GFP (1/1000, Aves GFP-1020), rabbit anti ds-red (1/1000, Clontech 632496), and rat anti-LAMP1 (1/500, BD Biosciences 553792) when applicable. Cells were washed three times in PBST and incubated for 2 h in PBST with goat anti-chicken antibodies coupled to AlexaFluor 488 (1/2000, Molecular Probes A11039), goat anti-rabbit coupled to AlexaFluor 594 (1/2000, Molecular Probes A11012), and goat anti-rat coupled to DyLight 650 (1/500, Invitrogen SA5-100021). After three washes in PBST, nuclei were stained with 4,6′-diamidino-2-phenylindole (DAPI) (1 µg/mL in PBS) for 5 min. Samples were mounted with ProLong™ Gold Antifade mounting medium (Molecular Probes) and kept at −20°, protected from light, until confocal imaging.

### 4.6. Image Acquisition and Analysis

Confocal images were acquired (Leica SP5) using a 63× (Numerical aperture (NA) 1.4) oil immersion objective and analyzed with ICY software (http://icy.bioimageanalysis.org/) as previously described [28]. Briefly, quantification was performed on a single-plane image from a *z*-stack within two sequential steps. First, the plasma membrane and cytoplasmic compartment were defined for each neuron. Each neuron was carefully delineated using the “free-hand area” tool. This initial Region of interest (ROI) was filled with the “fill holes in ROI” plugin to define the total cell area (ROI _total_). ROIs were then processed to generate two ROIs corresponding to the cell periphery and the cytoplasm. On the basis of staining in basal conditions, we estimated that most of the plasma membrane staining was found over an 8 pixel thickness. Therefore, we automatically eroded, with the “Erode ROI” plugin, the ROI _total_ by 8 pixels and subtracted this new ROI (ROI _cyto_) from ROI _total_ to obtain a ROI corresponding to the cell periphery (ROI _peri_).

The spots were then detected in each channel and the amount of co-localization determined in each region of interest. To detect the specific signal in each ROI, we used the “spot detector” plugin which relies on the wavelet transform algorithm [43]. By carefully setting the sensitivity threshold and the scale of objects to detect, it allows the detection of spots even in images with low signal-to-noise ratio. In our conditions, the sensitivity threshold was fixed between 50 and 60, and the scale of objects was set at 2 (pixel size 3) for mu and delta receptors. Once parameters were defined, images were processed with the tool “protocol” in ICY, which is a graphical interface for automated image processing. Data including the number of spots detected in each channel and ROI, the number of co-localized objects, and the ROI area were automatically collected in excel files. Objects were considered co-localized if the distance of their centroid was equal to or less than 3 pixels. The protocol used in these analyses is available online (http://icy.bioimageanalysis.org/protocol/newcolocalizer-with-binary-and-excel-output-v1_batch/). To obtain histograms, we calculated object densities for each receptor reported to the surface of each ROI. Membrane-to-cytoplasm density ratios were calculated to illustrate the subcellular distribution of each receptor. The extent of co-localization was calculated according to the following formula for each ROI: $\%\;\mathrm{colocalization}=100\times (\frac{\mathrm{colocalized}\;\mathrm{mu}\;\mathrm{and}\;\mathrm{delta}\;\mathrm{objects}}{\sum (\mathrm{detected}\;\mathrm{mu}\;\mathrm{and}\;\mathrm{delta}\;\mathrm{objects})})$. The extent of internalization is expressed as the ratio of membrane/cytoplasm immunoreactivity densities for each receptor or co-localized mu–delta. Co-localization of the two receptors is expressed as the percentage of co-localized mu-mCherry and delta-eGFP signals reported to the total immunoreactivity.

### 4.7. Statistical Analysis

Statistical analyses were performed with Graphpad Prism V7 software (GraphPad, San Diego, CA, USA). Normality of the distributions and homogeneity of the variances were checked before statistical comparison to determine appropriate tests. One-way nonparametric (Kruskal–Wallis followed by Dunn’s multiple-comparison test) or parametric one-way ANOVA test (followed by Dunnett’s multiple-comparison test) were used to compare different experimental groups. A two-way ANOVA followed by post hoc Tukey’s test for multiple comparisons was used for multiple factor comparisons. Results in graphs and histograms are illustrated as means ± standard error of the mean (SEM).

## 5. Conclusions

Our data demonstrate for the first time that co-expression of native mu and delta opioid receptors in hippocampal neurons alters the intracellular fate of the mu opioid receptor in a ligand-selective manner. This observation supports functional heteromerization of the two receptors that would contribute to the fine-tuning of mu opioid receptor signaling. It, therefore, highlights an interesting emerging concept for the development of novel therapeutic drugs and strategies. Importantly, our study also emphasizes the need to perform pharmacological studies on native receptors due to the limited translational value of data collected in co-transfected cells.

## Figures and Tables

**Figure 1 molecules-25-04493-f001:**
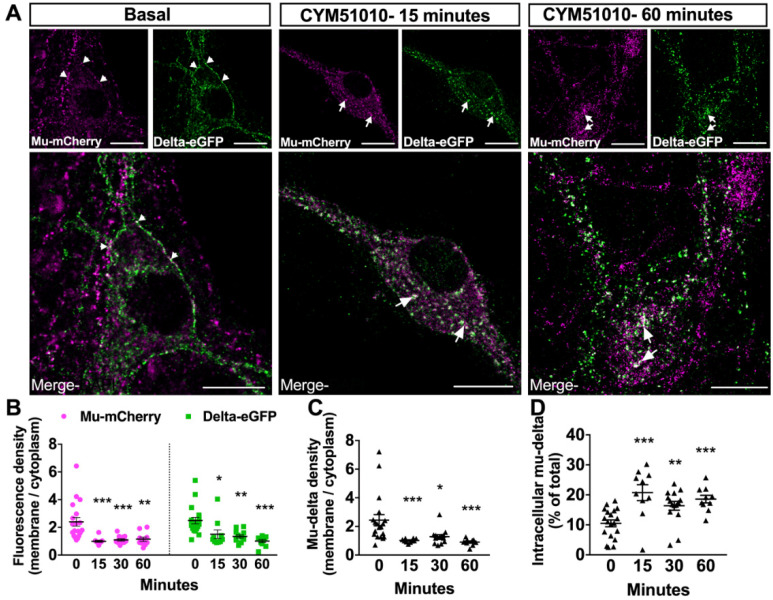
Mu and delta opioid receptors co-internalize upon CYM51010 activation in primary hippocampal cultures. (**A**) Representative confocal images showing mu-mCherry and delta-eGFP fluorescence localized at the plasma membrane (arrowheads) under basal condition or internalized in vesicle-like structures 15 or 60 min after CYM51010 (400 nM) application (arrows). Scale bar = 10 μm. (**B**) Receptor internalization induced by CYM51010 application (400 nM) expressed as a ratio of membrane-associated versus intracellular fluorescence densities for each receptor. Two-way ANOVA F_treatment_ (3, 94) = 17.98; *p* < 0.0001. F_receptor_ (1, 94) = 1.06; F_interaction_ (3, 94) = 0.54. Tukey’s post hoc test for mu-mCherry, *** *p* < 0.001, ** *p* = 0.01. Tukey’s post hoc test for delta-eGFP, * *p* = 0.02, ** *p* = 0.002, *** *p* < 0.001; *n* = 10 to 20 neurons per group from at least three independent cultures. (**C**) Subcellular redistribution of mu–delta heteromers expressed as a ratio of membrane-associated versus intracellular fluorescence densities for co-localized mu-mCherry and delta-eGFP receptors. One-way ANOVA (F (3, 48) = 13.64; *p* < 0.0001) followed by multiple-comparison Dunn’s post hoc test. * *p* = 0.03, *** *p* < 0.001; *n* = 10–20 neurons per group from at least three independent cultures. (**D**) Fraction of cytoplasmic mu-delta heteromers expressed as the percentage of mu-mCherry and delta-eGFP overlapping objects detected in vesicle-like structures at the different times. Kruskal Wallis test (*p* < 0.0001) followed with multiple comparisons Dunn’s test. ** *p* < 0.01, 30 min vs basal, *** *p* < 0.001 15 min and 60 min vs basal. N = 10 to 20 neurons per group from at least 3 independent cultures.

**Figure 2 molecules-25-04493-f002:**
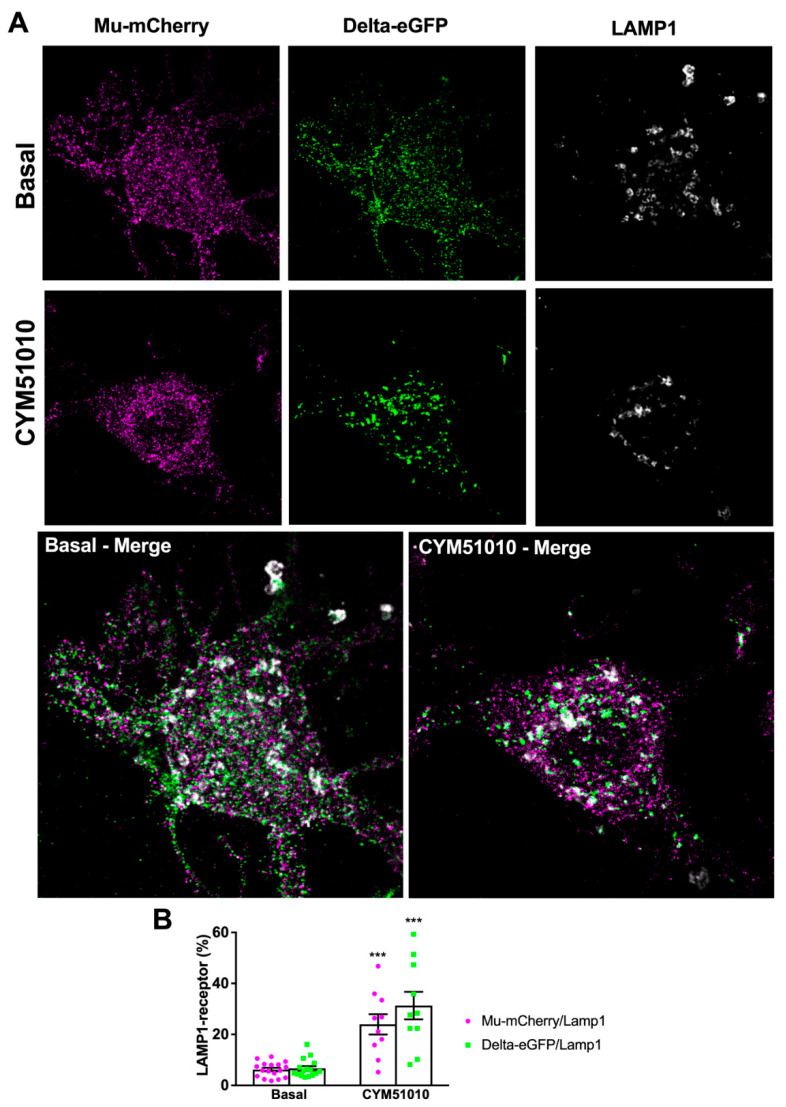
Mu and delta opioid receptors co-localize in the lysosomal compartment upon CYM51010 activation in primary hippocampal cultures. (**A**) Representative confocal images showing mu-mCherry–delta-eGFP colocalization with LAMP1 immunoreactive compartment under basal conditions or 60 min after CYM51010 application (400 nM). Scale bar = 10 μm (inset scale bar = 2.5 μm). (**B**) Drug treatment induces statistically significant increase in the amount of colocalization of mu-mCherry/delta-eGFP colocalization with LAMP1 labeling. Two-way ANOVA F_drug treatment_ (1, 49) = 62.70; *p* < 0.0001. F_receptor_ (1, 49) = 2.12, *p* = 0.15; F_interaction_ (1, 49) = 3.65, *p* = 0.2. Tukey’s post hoc test: *** *p* < 0.001 for both mu-mCherry and delta-eGFP; *n* = 10–20 neurons per group from at least three independent cultures.

**Figure 3 molecules-25-04493-f003:**
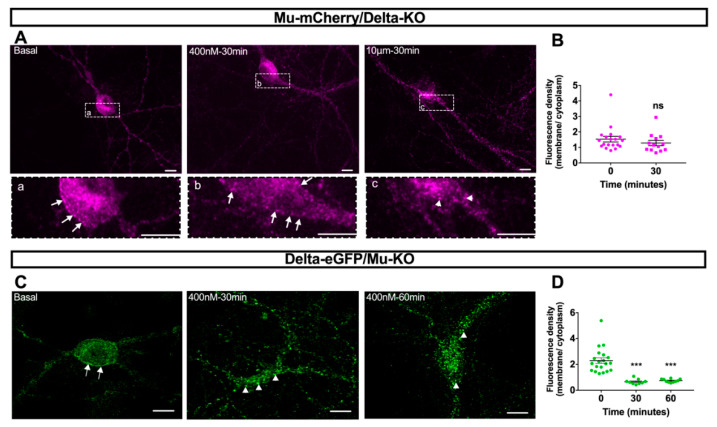
CYM51010 internalization of mu or delta opioid receptors in primary hippocampal cultures from mice deficient for one of the receptors. (**A**) Representative confocal images showing that mu-mCherry is associated with the plasma membrane (arrows) in basal conditions and 30 min after CYM51010 (400 nM) addition in delta-knockout (KO) mice. Scale bar = 10 μm. (**B**) Mu-mCherry internalization induced by CYM51010 application expressed as a ratio of membrane-associated versus intracellular fluorescence densities. Mann–Whitney test, *p* = 0.20; *n* = 13 to 20 neurons per group from at least three independent cultures. (**C**) Representative confocal images showing that delta-eGFP is predominantly associated with the plasma membrane in basal conditions (arrows) in mu-KO mice, whereas the association is mostly intracellular at 30 and 60 min after CYM51010 (400 nM) addition (arrowheads). Scale bar = 10 μm. (**D**) Delta-eGFP internalization induced by CYM51010 application expressed as a ratio of membrane-associated versus intracellular fluorescence densities. Kruskal–Wallis test (*p* < 0.0001) followed by Dunn’s multiple comparison test. Significant differences after multiple-comparison tests are expressed as *p* < 0.001 (***) compared to basal group; *n* = 9–20 neurons per group from at least three independent cultures.

**Figure 4 molecules-25-04493-f004:**
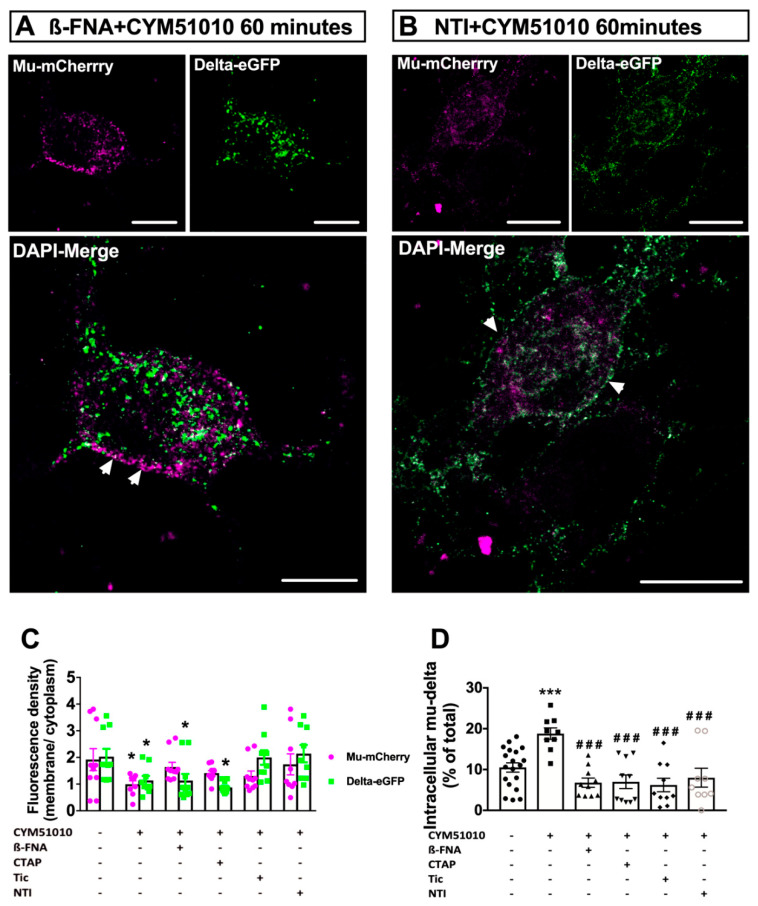
Antagonist pretreatment abolishes mu–delta opioid receptor co-internalization by CYM51010 in primary hippocampal cultures. (**A**) Representative confocal images showing mu-mCherry predominant localization at the plasma membrane and delta-eGFP extensive internalization after pretreatment with the mu antagonist β-FNA (200 nM) for 15 min, followed by incubation for 60 min with CYM51010 (400 nM). Scale bar = 10 μm. (**B**) Representative confocal images showing mu-mCherry and delta-eGFP predominant localization at the plasma membrane after pretreatment with delta antagonist naltrindole (200 nM) (NTI) for 15 min, followed by incubation for 60 min with CYM51010 (400 nM). Scale bar = 10 μm. (**C**) Pretreatment with the mu antagonists β-FNA or CTAP (200 nM) blocks mu-mCherry but not delta-eGFP internalization, whereas pretreatment with the delta antagonists naltrindole (NTI) and tic-deltorphin (tic) (200 nM) prevent internalization of both mu-mCherry and delta-eGFP. Receptor internalization is expressed as a ratio of membrane-associated versus intracellular fluorescence densities for each receptor. Two-way ANOVA F _treatment_ (5, 104) = 4.73, *p* = 0.0001. F_receptor_ (1, 104) = 0.1, *p* = 0.84; F_interaction_ (5, 100) = 1.96; *p* = 0.0006. Multiple comparisons with Tukey’s post hoc test, * *p* = 0.04 basal vs. CYM51010 for mu-mCherry, * *p* = 0.04 basal vs. CYM51010, * *p* = 0.04 basal vs. β-FNA, * *p* = 0.04 basal vs. CTAP; *n* = 9–20 neurons per group from at least three independent cultures. (**D**) Mu-mCherry/delta-eGFP co-internalization is prevented by treatment with either mu or delta antagonists. Percentage of colocalized receptors in the cytoplasm after drug treatment. The fraction of cytoplasmic mu–delta heteromers is expressed as the percentage of mu-mCherry and delta-eGFP overlapping objects detected in vesicle-like structures 60 min after CYM51010 application. One-way ANOVA (*p* < 0.0001) followed by multiple-comparison Dunnett’s test. Significant differences after multiple comparisons tests are expressed as *** *p* < 0.001 when compared to basal group and ^###^
*p* < 0.001 when compared to CYM51010 without antagonists; *n* = 9–20 neurons per group from at least three independent cultures.

**Figure 5 molecules-25-04493-f005:**
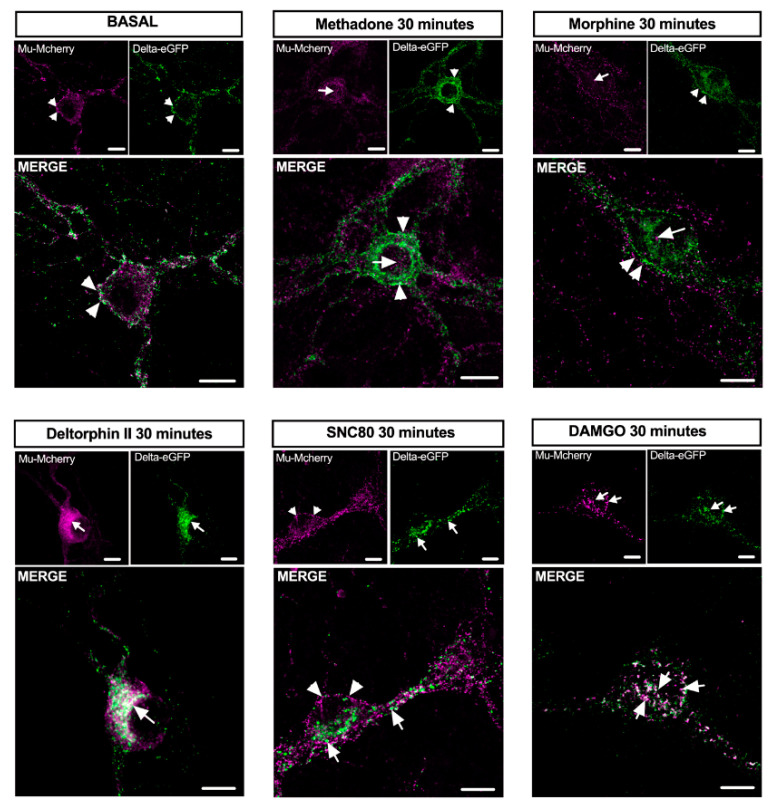
Mu–delta opioid receptor co-internalization is ligand-selective in primary hippocampal cultures. Representative confocal images showing mu-mCherry and delta-eGFP fluorescence at the plasma membrane (arrowheads) under basal conditions or co-internalized in vesicle-like structures (arrows) 30 min after DAMGO (1 μM) or deltorphin II (100 nM), but not SNC80 (100 nM), morphine (10 μM), or methadone (1 μM). Scale bar = 10 μm.

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
