# Peer review of "Heteromerization of Endogenous Mu and Delta Opioid Receptors Induces Ligand-Selective Co-Targeting to Lysosomes"

_molecules, 2020, doi:10.3390/molecules25194493_

Round 1
Reviewer 1 Report
General Questions/Comments:
- Have you quantified the expression of the fluorescent tagged MOR and DOR and compared it to the expression of native receptors? Please report these values.
- I know the red/green color scheme is traditional for these tags, but it makes it impossible for red/green colorblind people to read your figures. Please consider making ADA compliant.
- 3. Elaborate on why CYM51010 would produce DOR internaliztion in MOR KO mice if CYM51010 is MORDOR hetermomer specific. Doesn't this call into question the specificity of this tool and therefore it's usefulness? In the original CYM51010 paper MOR atibodies, DOR antibodies, and MORDOR antibodies all blocked CYM51010 activity (https://pubmed.ncbi.nlm.nih.gov/23818586/). An alternate interpretation is that CYM51010 interacts with both MOR and DOR, not necessarily the heteromer. Please address this in the discussion.
- Check consistency of "CYM51010" vs "CYM 51010"
- Why were different ligands used for the HEK experiments than used in cultured neurons?
- Highlight what this paper contributes to the field of MORDOR heteromers. I am not sure what the new take away information is, you need to emphasize your point more clearly.
Specific Questions/Comments:
- In figure 4C make all asterisks to denote significance black (one is green)
- Fix beta symbol line 254 and 265 and 296
Author Response
We would like to thank the reviewer for the comments and suggestions. We took them into account as best as possible.
Expression level of the mu opioid receptor in the mouse hippocampus was estimated to 10-15 fmol/mg protein [1, 2] and was not significantly affected by the fusion to the mcherry fluorescent protein [3]. Delta opioid receptor expression level was estimated to 30-50 fmol/mg protein in the mouse hippocampus [1, 2, 4, 5] but was increased about 2-fold in the case of the fluorescent receptor fusion [6].
This information is now included in the manuscript lines 241-242 and lines 245-246
We would like to apologize for not considering this aspect. We changed the LUT from red to magenta in all figures.
We agree with the reviewer that CYM51010 is not specific for mu-delta and also binds to delta or mu opioid receptors when expressed alone. Several results indeed indicate an interaction with the mu opioid receptor. In the original article referred to by the reviewer, mu-delta antibody and, to a lower extent, mu antibody decreased [35S]GTPgS activation by CYM51010 [7]. The recent study by Tiwari at al 2020 [8] also indicates that CYM51010 has analgesic properties in delta knockout mice. Our own unpublished data indicate that CYM51010 is analgesic in delta knockout animals even if the ligand poorly internalizes the mu receptor in the absence of the delta receptor (shown in figure 3). Interaction with the delta receptor was not previously reported as [35S]GTPgS activation by CYM51010 was not decreased by delta antibody [7] and no effect was observed on mechanical or thermal allodynia in neuropathic mu knock out animals [8]. However, our study clearly points to internalization of the delta opioid in the absence of mu receptors (shown in figure 3). Moreover, our preliminary unpublished data suggest that CYM51010 can modify the thermal nociceptive threshold in mu knock-out mice. Altogether, this limits the interest of the compound and requires the development of more specific ones. This is now discussed lines 263-282.
Although CYM51010 binds to both mu and delta receptors and poorly internalizes mu opioid receptors in delta opioid knock out mice, it induces co-internalization when the two receptors are co-expressed. We would however like to stress out that these results support the hypothesis of a different conformation of the mu receptor following CYM51010 activation depending of the presence or not of the delta opioid receptor.
We have used CYM51010 everywhere
The existence of mu-delta heteromers in vivo is still debated. The absence of mu-delta co-internalisation by the delta agonist SNC80 in the spinal cord [9], a ligand that co-internalizes the two receptors in co-transfected cells, has been put forward as an argument against the presence of endogenous mu-delta heteromers. This outlines the need to test the impact of ligands in native context. All experiments presented in the manuscript were performed in primary hippocampal neurons but we have tested several ligands that were previously reported as co-internalizing the receptors in transfected cells to determine their translational value. Our study clearly indicates that mu-delta co-internalization is ligand dependent and that data obtained in co-transfected cells cannot be systematically extrapolated to native contexts.
As mentioned in the abstract, our results establish for the first time that neuronal co-expression of native mu-delta opioid receptors changes the intracellular fate of the mu opioid receptor in a ligand dependent manner. It also shows that the co-internalization is driven by the delta opioid receptor and requires both receptors to be in an active conformation. It also demonstrates that the translational value of data collected on co-transfected cells remains limited likely due to differences in the cellular content. The discussion and conclusion have been revised to highlight these aspects.
This has been modified
This has been done
References
- Kitchen, I., et al., Quantitative autoradiographic mapping of mu-, delta- and kappa-opioid receptors in knockout mice lacking the mu-opioid receptor gene. Brain Res, 1997. 778(1): p. 73-88.
- Lesscher, H.M., et al., Receptor-selective changes in mu-, delta- and kappa-opioid receptors after chronic naltrexone treatment in mice. Eur J Neurosci, 2003. 17(5): p. 1006-12.
- Erbs, E., et al., A mu-delta opioid receptor brain atlas reveals neuronal co-occurrence in subcortical networks. Brain Struct Funct, 2015. 220(2): p. 677-702.
- Goody, R.J., et al., Quantitative autoradiographic mapping of opioid receptors in the brain of delta-opioid receptor gene knockout mice. Brain Res, 2002. 945(1): p. 9-19.
- Chu Sin Chung, P., et al., A novel anxiogenic role for the delta opioid receptor expressed in GABAergic forebrain neurons. Biol Psychiatry, 2015. 77(4): p. 404-15.
- Scherrer, G., et al., Knockin mice expressing fluorescent delta-opioid receptors uncover G protein-coupled receptor dynamics in vivo. Proc Natl Acad Sci U S A, 2006. 103(25): p. 9691-6.
- Gomes, I., et al., Identification of a mu-delta opioid receptor heteromer-biased agonist with antinociceptive activity. Proc Natl Acad Sci U S A, 2013. 110(29): p. 12072-7.
- Tiwari, V., et al., Activation of micro-delta opioid receptor heteromers inhibits neuropathic pain behavior in rodents. Pain, 2020. 161(4): p. 842-855.
- Wang, D., et al., Functional Divergence of Delta and Mu Opioid Receptor Organization in CNS Pain Circuits. Neuron, 2018. 98(1): p. 90-108 e5.

Reviewer 2 Report
Review of “Heteromerization of endogenous mu and delta opioid receptors induces ligand selective co-targeting to lysosomes.”
Summary:
The manuscript focuses on evaluating how the heteromerization of endogenous mu and delta opioid receptors alters agonist-induced internalization and trafficking. The manuscript is interesting and nicely leverages endogenously tagged receptors to evaluate agonist induced trafficking. The results are however somewhat limited due to the sole experimentation centered around fixed cell immunofluorescence with limited co-staining of downstream endocytic compartments. Additionally, many of the results are difficult to interpret due to a lack of real-time kinetic data for the receptor trafficking, and the need to concretely show that the receptors are in fact on the surface and heterodimerized. The manuscript overall is of interest to the field but seems somewhat preliminary with lack of additional data on effects of heteromerization signaling, detailed trafficking pathway, mechanism of internalization, use of live cell imaging, and fate of receptor complexes following internalization. The work would be much improved following the authors can address several major concerns:
Major Comments:
- The authors are claiming that most colocalization and potentially heteromerization occurs at the plasma membrane. With endogenous expression it is often difficult to segment the plasma membrane from cytoplasmic regions due to the punctate staining as shown in these figures. These receptors could be at the surface or in vesicles docked near the surface. The paper would be strengthened by costaining with a membrane marker to definitively show the population of receptors on the cell surface. Alternatively, the authors could antibody stain for the receptor expression without permeabilization to get a ratio of membrane to internal receptor pool.
Similarly, the authors state that quantification of receptor membrane density and colocalization is performed using ICY software by defining ROIs of the membrane and cytoplasm. How are these ROIs being determined? Is this arbitrary or are the authors using a membrane marker? It seems that the experiments were performed on fixed samples which should allow for membrane staining and non-biased ROI determination.
- The authors state in line 89-90 that colocalization at the plasma membrane indicate being in close physical proximity and suggests constitutive heteromerization. While this could be true, it also seems just as likely that the receptors could be within similar membrane domains and or within the same vesicles but not necessarily in a physical heteromer. Additional biochemical evidence of heteromerization at the plasma membrane should be performed to confirm that in these neurons the receptors are physically associating in addition to citing the previous literature which did not determine where within the cell the heteromers were present.
- All experiments appear to be evaluated with fixed cell fluorescence imaging. A benefit of having endogenously fluorescently labeled receptors is for live imaging capabilities. Many of the conclusion in the paper would be strengthened with the addition of live imaging showing the internalization from the membrane overtime of both the mu and delta receptors.
- The colocalization in Figure 2 is hard to evaluate. Due to saturation of the fluorescence signal and abundance of the receptor expression throughout the cytoplasm higher resolution images are needed to accurately determine if the receptors are in fact colocalizing with LAMP1. Additionally, these conclusions would be strengthened by showing the time course of colocalization with LAMP1, as earlier time points should have low colocalization and the later 60 min time point should show increased colocalization with LAMP1. As an alternative approach, these experiments could be performed live using a lysotracker dye to visualize the kinetics over time within the same cell.
- In Figure 3 the authors are concluding that mu internalization and trafficking to late endocytic compartments by CYM51010 is dependent upon the delta receptor expression. The authors did see mu internalization at higher concentrations of agonist suggesting delta receptor role could be to increase agonist efficacy and not a dependence on receptor expression. Further, what endocytic vesicles are the mu receptors in without delta present? Additional costaining for lysosomal and endocytic compartments are needed to support the authors conclusions.
- The data in Figure 4 is difficult to interpret without additional co-staining from endocytic compartments and the plasma membrane. While the authors interpretation is plausible, an alternative interpretation is that with the addition of the mu selective antagonist the mu and delta receptors are sorting into different endocytic pathways. Additionally, using the delta selective antagonists the delta receptors do appear more surface localized; however, the mu receptors have a distinct and more cytoplasmic distribution. Co-staining with early endocytic, late endocytic, and lysosomal markers would clarify this to support the authors conclusion. Quantification of each receptor individually and their distribution is also needed for this figure.
- The data showing differences between multiple agonists in Figure 5 is very interesting. The authors state that in some cases mu (DAMGO) or delta (Delt II) agonists drive internalization, but with other selective agonists they do not see co-internalization. How are the authors ruling out the possibility that DAMGO and Delt II could be activating both receptors independently at the concentrations used? Similar to other data shown by the authors, these conclusions are somewhat limited by only performing fixed cell single time point imaging. To clearly demonstrate this point, live imaging of the fluorescence distribution over time is needed. Alternatively, membrane and internal compartmental markers with image quantification would support these conclusions.
Minor Comments:
- The overall writing and clarity of the introduction needs improvement. Discussion on the neuronal cell types co-expressing mu and delta should be added to the introduction, as well as the previously reported internal distribution for the delta receptor. While overall content is acceptable the statements made read very disjointed jumping from one to another. Additionally, please make sure to cite all statements in the introduction. In several instances the claims made are not supported by references and there are plenty of reviews and primary article to cite from these statements.
- Image LUT intensity needs to be more consistent. In several figures the mu or delta images are difficult to see. One option to improve the contrast could be to convert the single channel images to black and white and have only the merged image in color. The images are also not color-blind sensitive and the authors should consider converting to a more appropriate magenta vs red before publication.
Author Response
Reviewer 2
We would like to thank the reviewer for their comments and suggestions. We took them into account as best as possible. please see attached document

Reviewer 3 Report
The manuscript “Heteromerization of endogenous mu and delta opioid receptors induces ligand selective co-targeting to lysosomes” by Derouiche et all makes use of mice co-expressing delta-eGFP with mu-mCherry to investigate receptor internalization in primary hippocampal cultures. They observe that some ligands (CYM51010, DAMGO, deltorphin II) but not others (morphine, methadone, SNC80) induce mu-delta co-internalization. They find that mu-delta co-internalization is driven by the delta opioid receptors and requires that both receptors be in their active conformation. Moreover, they find that the co-internalized receptors are targeted to the lysosomal compartment. This is a well-conducted set of studies with appropriate use of controls that justify their conclusions.
A few clarifications would improve the manuscript.
- In Fig 4C, on what basis where the antagonist concentrations chosen? Also, under CYM51010+CTAP why is the asterix in green?
- In Fig 5, a quantitation of the data for the different panels should be included similar to what was done in figs 1-4.
- It is not clear what was the basis for the selection of the doses of agonists used in Fig 5. Were dose response studies done?
- On page 12 line 309 the authors state “ or in 12-well plates coated with poly-lysine for ERK phosphorylation studies”. There are no ERK phosphorylation studies in this manuscript
- On page10 line 256 authors write “mu agonist CTOP” - this should be mu antagonist.
- On page 2 line 55 “assorted” should be changed to “associated”.
Author Response
We would like to thank the reviewer for their comments and suggestions. We took them into account as best as possible.
The concentrations of antagonists were determined as the lowest ones blocking co-internalization based on a dose response curve (0-1microM)
The color of the asterix was a mistake. It is now black similarly to all others
The first step of our study consisted in testing a broad range of mu and delta agonists at various concentrations ranging from 1 nM to 10 microM in cultures expressing the two receptors and in cultures expressing one receptor and deficient for the other. These initial kinetics were performed on 2-3 independent cultures but co-internalization was only qualitatively assessed using a limited number of confocal acquisitions. The latter is unfortunately too low to perform statistically valid quantification as quantification of the internalization requires the imaging of 10-30 neurons per culture.
For each ligand, we first performed kinetics at different concentrations ranging from 1 nM to 10 microM in cultures expressing the two receptors and in cultures expressing only one receptor and deficient for the other. The broad range was chosen because the reported Ki affinities for each receptor vary from the nanomolar (e.g. SNC80 for the delta receptor) to 100 nanomolar (e.g DAMGO for the mu receptor) range and we did not know which concentrations would promote internalization. Also, we did not know whether co-internalization would be induced by in a different concentration of agonist compared to internalization of each receptor alone. Following this initial screen, we used the lowest agonist concentrations giving consistent internalization across experiments.
We would like to apologize. This refers to results that are not included in this manuscript and has been removed (now line 331).
We would like to apologize for the mistake. It is now corrected
This has been modified
Round 2
Reviewer 1 Report
1. Make sure colors in the text describing figures (red/green) match the colors in the figures (magenta/green)
Reviewer 2 Report
Thank you for clarifying the analysis methods, and additions to the discussion. The additional detail is helpful. While the reviewer agrees that it can be difficult to investigate the trafficking pathway in primary neurons, and to perform live imaging due to endogenous expression levels, it would make the manuscript stronger to include these data. In any case, possibly including some discussion on why this was not performed or what future directions would be could address this limitation. More discussion should be added about how the C-terminal fluorescent tag on the mu-opioid receptor could interfere with required trafficking motifs that might result in a decrease or altered receptor recycling. Minor Concerns: Missing words in line 230: "Also, we did not evidence mu-delta co231 internalization by the mu agonist methadone although this ligand induced mu-delta co-trafficking in 232 in co-transfected cells [19]." The wording of this sentence implies that the manuscript demonstrated that a conformational change is occurring and leading to altered beta-arrestin recruitment. Was this data shown in this manuscript? If not, please alter the statement to reflect that this is a hypothesis derrived from the results. Discussion line 275: "On the opposite, CYM51010 induced internalization of the mu opioid receptor when associated with the delta opioid receptor suggesting that its binding triggered a different conformation of the mu opioid receptor that allowed beta-arrestin recruitment."